# An Ontological Clinical Decision Support System Based on Clinical Guidelines for Diabetes Patients in Sri Lanka

**DOI:** 10.3390/healthcare8040573

**Published:** 2020-12-18

**Authors:** Sajith Madhusanka, Anusha Walisadeera, Gilmini Dantanarayana, Jeevani Goonetillake, Athula Ginige

**Affiliations:** 1Department of Computer Science, Faculty of Science, University of Ruhuna, Matara 81000, Sri Lanka; kds.madhusanka@gmail.com (S.M.); gilmini@dcs.ruh.ac.lk (G.D.); 2Department of Information Systems Engineering, University of Colombo School of Computing, University of Colombo, Colombo 00700, Sri Lanka; jsg@ucsc.cmb.ac.lk; 3School of Computer, Data and Mathematical Sciences, Western Sydney University, Sydney 2751, Australia; A.Ginige@westernsydney.edu.au

**Keywords:** clinical guidelines, clinical decision support system, ontology, diabetes, healthcare

## Abstract

Health professionals should follow the clinical guidelines to decrease healthcare costs to avoid unnecessary testing and to minimize the variations among healthcare providers. In addition, this will minimize the mistakes in diagnosis and treatment processes. To this end, it is possible to use Clinical Decision Support Systems that implement the clinical guidelines. Clinical guidelines published by international associations are not suitable for developing countries such as Sri Lanka, due to the economic background, lack of resources, and unavailability of some laboratory tests. Hence, a set of clinical guidelines has been formulated based on the various published international professional organizations from a Sri Lankan context. Furthermore, these guidelines are usually presented in non-computer-interpretable narrative text or non-executable flow chart formats. In order to fill this gap, this research study finds a suitable approach to represent/organize the clinical guidelines in a Sri Lankan context that is suitable to be used in a clinical decision support system. To this end, we introduced a novel approach which is an ontological model based on the clinical guidelines. As it is revealed that there are 4 million diabetes patients in Sri Lanka, which is approximately twenty percent of the total population, we used diabetes-related guidelines in this research. Firstly, conceptual models were designed to map the acquired diabetes-related clinical guidelines using Business Process Model and Notation 2.0. Two models were designed in mapping the diagnosis process of Type 1 and Type 2 Diabetes, and Gestational diabetes. Furthermore, several conceptual models were designed to map the treatment plans in guidelines by using flowcharting. These designs were validated by domain experts by using questionnaires. Grüninger and Fox’s method was used to design and evaluate the ontology based on the designed conceptual models. Domain experts’ feedback and several real-life diabetic scenarios were used to validate and evaluate the developed ontology. The evaluation results show that all suggested answers based on the proposed ontological model are accurate and well addressed with respect to the real-world scenarios. A clinical decision support system was implemented based on the ontological knowledge base using the Jena Framework, and this system can be used to access the diabetic information and knowledge in the Sri Lankan context. However, this contribution is not limited to diabetes or a local context, and can be applied to any disease or any context.

## 1. Introduction

Healthcare is the improvement or maintenance of health through the prevention, diagnosis, and treatment of people’s injuries, illnesses, and other physical and mental disorders provided by allied health professionals. The health professionals use their expert knowledge to provide the best possible healthcare services at a low cost. Still, many healthcare systems face increased healthcare costs due to an increased demand for care, expensive technologies, increased aging population, and variations in the standard of healthcare service delivery among providers, hospitals, and in different geographical regions. Some of these reasons arise from improper care, overuse of services, underuse of services, the inherent desire of healthcare professionals to provide the best possible care, and also the intrinsic desire of a patient to seek the best possible care [1]. The main reason for these variations in the healthcare service is due to decisions made by health professionals, which depend on their expertise level. A health professional with a high expertise level makes better decisions regarding diagnosis and treatments than a health professional with a lower expertise level.

In this study, we focused on only the clinical guidelines related to diabetes management. Diabetes is a chronic disease that occurs either when the pancreas does not produce enough insulin or when the body cannot effectively use the insulin it produces. Insulin is a hormone that regulates blood sugar. Hyperglycemia, or raised blood sugar, is a common effect of uncontrolled diabetes and over time leads to serious damage to many of the body’s systems, especially the nerves and blood vessels. According to the World Health Organization, in 2014, 8.5% of adults aged 18 years and older had diabetes [2]. In 2016, diabetes was the direct cause of 1.6 million deaths, and in 2012, high blood glucose was the cause of another 2.2 million deaths [2]. There are 4 million diabetes patients in Sri Lanka [3]. This is approximately twenty percent of the total population–that is, one in five people have diabetes in Sri Lanka. It has been identified that diabetes will be the fifth main cause of deaths around the world in 2030 [2]. According to the World Health Organization, it is currently the seventh main cause of deaths around the world. According to the International Diabetes Federation, in 2017, approximately 425 million adults (20–79 years) were living with diabetes, and by 2045 this will rise to 700 million [4]. Around one fifth of them live in the Eastern and Southern Asia Pacific regions; therefore, as Sri Lankans, we will face many problems in the near future. This will affect the economic development of the country as well as the productivity of the individual. This challenge can be achieved by providing a proper health education. In order to properly manage diabetes, we must establish good healthcare in the country and thus clinicians must implement clinical guidelines effectively.

In the last decade, clinical guidelines have become an increasingly familiar part of clinical practice [1]. According to the Institute of Medicine, clinical guidelines are defined as: “systematically developed statements to assist practitioner and patient decisions about appropriate healthcare for specific clinical circumstances” [5]. Clinical guidelines provide information on what diagnoses or tests should be performed, how medical or surgical treatment completed, how long the patient stays in the hospital, or other information about clinical practice. Health professionals should follow the clinical guidelines to decrease the healthcare cost to avoid unnecessary testing, and to minimize the variations among healthcare providers. Furthermore, it is possible to minimize the mistakes in the diagnosis and treatments and thus in turn standardize healthcare service delivery. However, most guidelines are usually presented in non-computer-interpretable narrative text or non-executable flow chart formats. These non-computable models limit the usefulness of the guidelines, as the knowledge contained in the guidelines is not readily available when treating the patients [6]. However, healthcare organizations generally focus more on developing guidelines than implementing guidelines for routine use in clinical settings [7]. Therefore, it is important to find the most suitable approach to implement clinical guidelines, and this study mainly focused on designing a formal knowledge representation model for this purpose.

Several studies have shown that the best strategy for implementing clinical guidelines is a computer-based clinical decision support system which is supported to integrate patient-specific support and decision-making into clinical workflows and analyze patients’ feedback to improve the quality of the decision-making process [8,9]. There is substantial evidence that computer-based clinical decision support systems can positively impact healthcare providers [10,11]. However, computer-based clinical decision support systems have not been widely used outside of medical centers, and their impact on patient outcomes is marginal [12,13] because of the lack of standard knowledge representation models and data models to represent clinical guidelines [14]. To solve this problem, studies have explored the development of a standard model for guideline representation [15,16].

A general model for clinical guideline representation is an authoring tool for producing guidelines with insight into the clinical care process addressed by clinical guidelines and thus providing general knowledge for conceptual modeling [16]. It can be used as a basis for adaptation and integration into unequal systems [15] and can be used to identify the requirements for adapting guidelines to the user’s level of expertise in decision making [17]. A standard model can contribute significantly to the development of guidelines and facilitates knowledge management to implement useful guidelines [18]. It can also be used to promote standardized approaches to guideline dissemination [19]. Several process-oriented representation languages have proposed creating a standard format, such as GLIF, GUIDE, Asbru, EON, PRODIGY, and PROforma [20]. Although these languages provide a complete governance flow structure for describing procedural knowledge on healthcare processes, most of them have been designed with limited expressivity of institution-specific organizational knowledge and domain-specific medical knowledge. They are equally crucial for clinical practice. The guidelines are often a very complex and exist within a changeable clinical context [21]. Several studies suggested that ontology is an appropriate way to demonstrate clinical guidelines, which can be used to extend and represent different levels of clinical guidance, which are scalable, flexible, and expressive enough to define the complexity and the changing guidelines of the application domain [22]. The ontological model will thus overcome the limitations of the methods mentioned above.

Driven by the aforementioned factors, the study discussed in this paper proposes an ontological model to use in a clinical decision support system for diabetes patients especially in Sri Lanka. Furthermore, a novel approach is introduced to design an ontology that is based on the clinical guidelines.

The main contributions of this research work are as follows:Designing the conceptual models by using the Business Process Model and Notation 2.0 (BPMN 2.0) to map the clinical guidelines to increase the understandability and to validate the acquired knowledge from the clinical guidelines.Introducing a novel approach to design an ontology to represent the diabetes-related knowledge that is based on the conceptual models designed above based on BPMN by getting feedback/comments from domain experts.

This research is an original work on the knowledge representation of diabetes-related clinical guidelines and mainly focuses on designing a formal model to represent knowledge in the healthcare domain. The remainder of this paper is organized as follows. The methodology followed and conceptual designs and implementation architectures are discussed in Section 2. Section 3 elaborates on the evaluation phase according to different metrics in terms of validity and usability. Finally, the discussion, conclusion and future works are presented in Section 4.

## 2. Materials and Methods

Since the main objective of this study was to design the artifacts to represent the healthcare domain to increase the quality of the decision-making process, Design Science Research (DSR) was chosen as the research methodology. DSR is effective and consistent with the information system (IS) paradigm of designing and evaluating IT artifacts to increase problem-solving capabilities. The existing knowledge, theories, and methods are used to design and evaluate an artifact to solve the real-world problems. There are seven guidelines to follow with regard to the DSR paradigm [23]. These guidelines were followed in conducting this research. The conceptual model designing and implementation procedures of the proposed method are discussed in detail in Section 2.1 and Section 2.2, respectively.

### 2.1. Designing Conceptual Models

#### 2.1.1. Identify Diabetic-Related Clinical Guidelines

There are several clinical guidelines available for diabetes; among them, most are related to type 1 and type 2 diabetes. We used several locally developed clinical guidelines in this research because we focused on implementing clinical guidelines which can be used in the Sri Lankan context; they are:Clinical guidelines from the Ceylon College of PhysiciansClinical guidelines from the Sri Lanka College of Obstetricians and GynecologistsClinical guidelines from the Sri Lanka Diabetes Federation

These guidelines were developed from the existing guidelines published by various international professional organizations, including the American Diabetes Association, and modified to suit the local context. Local adaptability is essential to deciding on a management plan of diabetes when deciding treatments, as health professionals should consider the economic background of patients and their lifestyle. Therefore, we select the above clinical guidelines for implementing a model which can be used in a clinical decision support system (CDSS). We also studied clinical guidelines that were published by the American Diabetes Association and the World Health Organization.

#### 2.1.2. Map Clinical Guidelines into Conceptual Models

A conceptual model is essential in providing a better understanding of clinical guidelines for both domain experts and ontology developers. Models enable decision-makers to filter complexities that are not relevant in the real world and can focus on the most important parts of the system under study [24]. Since the clinical guidelines are process-oriented, we studied different methods that are used to model business processes. There are many popular methods and languages to model business processes, such as BPMN 2.0, event-driven process chain (EPC), and Unified Modeling Language (UML). Among them, BPMN is the most suitable business process modeling language for the healthcare domain, as it is the most suitable process modeling language to address the several role-related process modeling requirements of the healthcare domain [25]. BPMN is a well-defined enterprise modeling method to model the business process. Therefore, we used BPMN 2.0 as the method to design the conceptual model. We designed two conceptual models using BPMN, one to map the diagnosis process of type 1 and type 2, and the other one to map the diagnosis process of gestational diabetes. These conceptual models describe the general clinical pathway to diagnose type 1, type 2, and gestational diabetes. They include symptoms to consider, criteria to determine for screening, lab tests to perform, criteria to diagnose, and activities to carry out in the diagnosis of diabetes. We can use these models to identify concepts, relationships, and axioms (to represent the constraints) in the domain which are needed to design the ontology.

Figure 1 shows the screening and diagnosis of diabetes type 1 and type 2. This is based on diabetes management clinical practice guidelines which were published by the Sri Lanka College of Obstetricians and Gynaecologists. This diagram describes, step by step, how to screen, diagnose, and classify diabetes in patients. The main activities of the diagnosis process are numbered 1 to 10 in the diagram (see activities in Figure 1). Diabetes can be diagnosed on the basis of the A1C criteria or the plasma glucose criteria, with a fasting plasma glucose (FPG) or 2 h plasma glucose (2-h PG) 75 g oral glucose tolerance test (OGTT). The same tests are used to both screen and diagnose diabetes, and also these tests will detect individuals with prediabetes.

In Figure 1, the diagnosis process starts when the patient describes his/her symptoms and concerns. Identifying the symptoms of type 2 diabetes can be a challenge. Regular checkups with a doctor may find that blood sugar levels are elevated. The patients should clearly describe their symptoms and concerns because the clinicians take actions based on those symptoms and concerns. Most people believe that type 1 diabetes is only for children, but it can occur at any age. Clinicians may not be able to correctly diagnose type 1 diabetes in adults. They misunderstand the elevated glucose level for a sign of type 2 diabetes and recommend a new diet, exercise, and medication. After concerning patient symptoms, a comprehensive clinical assessment should be carried out at the first encounter of a patient with diabetes. During this assessment, clinicians can find out useful information about a patient’s lifestyle, behavioral, dietary, and pharmaceutical interventions. A detailed medical history, physical examination, and laboratory investigations should be obtained during the initial clinical assessment. By using these collected patient data, clinicians determine the need for screening for diabetes. For that, they should consider the criteria that are mentioned in the diagram (see number 3 activity note in Figure 1). The clinician should examine the patient’s blood test results to diagnose diabetes. There are standard cut off values for each diagnostic test to determine that the patient has diabetes or prediabetes.

Criteria for the diagnosis of diabetes:FPG > 126 mg/dL (7.0 mmol/L) OR2-h PG > 200 mg/dL (11.1 mmol/L) during an OGTT ORHbA1c > 6.5%

Criteria for the diagnosis of prediabetes:FPG 100–125 mg/dL (5.6–6.9 mmol/L) OR2-h PG 140–199 mg/dL (7.8–11.0 mmol/L) during an OGTT ORHbA1c 5.6–6.4%

Unless a clear clinical diagnosis (patient in a hyperglycemic crisis or with classic symptoms of hyperglycemia) is available, the diagnosis should be confirmed by repeating the same test with a new blood sample or by another test. Periodic testing for undiagnosed diabetes is recommended by measuring fasting plasma glucose according to the following schedule:Each year for people with impaired glucose tolerance (IGT) or impaired fasting glucose (IFG)Every 3 years for people at high risk with a negative screening blood test

Persons with risk factors who have a negative screening test are at risk of cardiovascular disease and the future development of type 2 diabetes, and thus should be given appropriate advice on risk factor reduction. Once diagnosed with diabetes, the clinician should classify patient diabetes as type 1 or type 2. To this end, they examine the patient’s response to oral medications as shown in the diagram (see number 9 activity in Figure 1). If the patient successfully responds to oral medications, then the patient is classified as a type 2 diabetes patient. If this is not the case, then the patient is classified as a type 1 diabetes patient. This is the way in which the diabetes diagnosis and classification process is carried out in the clinical environment. We designed these conceptual models to minimize the complexity of the clinical guidelines and increase the understandability.

Figure 2 shows the process of the diagnosis of gestational diabetes. It describes the screenings that are available for diagnosing gestational diabetes, the diagnosis criteria, and the other conditions that need to be considered when diagnosing, etc. Non-medical personnel can understand the flow of the diagnosis process without having a sound knowledge about the relevant clinical guidelines.

We also designed several flow charts to represent clinical guidelines related to diabetes management. When considering diabetes management, it is important to get a deep understanding of the structure of the diabetes management plans that are included in clinical guidelines. We need to highlight the key processing and decision points of diabetes management plans. Therefore, flowcharting is a simple modeling method to achieve those requirements. Then, we designed flowcharts to represent different diabetes management plans. Before modeling the clinical guidelines related to diabetes management, we classified the management plans into several categories, such as physical activities, non-insulin therapy, insulin therapy, management of cardiovascular risk factors, and complication management. Then, we designed separate flow charts for each category. Figure 3 shows the process of non-insulin therapy and the facts to be considered when deciding the non-insulin therapy strategy for a patient.

Before deciding the appropriate non-insulin therapy strategy for a patient, the clinician should decide a glycemic target according to the factors of the patient, such as age, duration of diabetes, and number of co-morbidities. After that, the clinician can recommend oral antibiotic agents according to the results of diabetes screenings (see Figure 3). If the value of diabetes screenings is higher than the glycemic target, then the clinician can decide to change the diabetes management plan into the insulin therapy strategy. According to Figure 3, it is important to monitor glycemic control monthly. These are the some of the important key processing and decision points included in the non-insulin therapy clinical guidelines. We mapped those important facts using flowcharting modeling, which makes it very easy to map the structure of the clinical guidelines. These models help us to decrease the complexity in the text format of the clinical guidelines. Therefore, the understandability of the diabetes management plans can be increased using these kinds of conceptual models.

#### 2.1.3. Ontology Design

When developing an ontology, the ontology design is a core aspect. However, there are no standard methods available to design an ontology. Therefore, selecting a suitable methodology to design an ontology depends on the nature of the application. There are different methods available such as Grüninger and Fox, METHONTOLOGY, and Uschold and King’s method that provide the guidance to design an ontology. We decided to use Grüninger and Fox’s method for designing the ontology because it provides a standard method to design as well as evaluate the ontology in a structured manner when compared with the other methods.

There are several steps to follow when designing an ontology by using this methodology. Firstly, we obtained use case scenarios in the management of diabetes mellitus done in [26] as motivating scenarios. Then, the informal competency questions were formulated for each scenario. Each motivating scenario was generalized to identify the concepts, sub-concepts, properties and axioms (criteria) for designing the ontology structure. For example, consider the following motivation scenario and corresponding informal competency questions.

Motivation Scenario 1:

A 46-year-old obese businessman (body mass index—32 kg/m^2^) with essential hypertension and type 2 diabetes mellitus (T2DM) of 8 years’ duration presented with poor glycemic control (glycated hemoglobin (HbA1c)—9.4%). He was on maximal dosage of metformin and sulfonylureas and has been following his diet and exercise schedule very strictly. He is also concerned about his weight and wants advice on which class of oral antidiabetic drug would be best suited in his case that may provide efficacious glycemic control, weight loss, and cardiovascular protection. He also wanted to know the side effects of these drugs and what measures he could follow to prevent these adverse events.

Informal Competency Question 1:


*A 46-year-old man with type 2 diabetes for eight years has cardiovascular risk factors such as hypertension and obesity (body mass index—32 kg/m^2^) and he has presented with poor glycemic control (glycated hemoglobin (HbA1c)—9.4%). He is on the maximum dose of metformin and sulfonylureas and has strictly followed his diet and exercise schedule. Which is the class of oral antibiotics that would provide effective glycemic control, weight loss, and cardiovascular safety?*


Informal Competency Question 2:


*What are the side effects of drugs which are best suited to provide efficacious glycemic control, weight loss, and cardiovascular protection and what measures can a patient follow to prevent these adverse events?*


Then, we identified key concepts in the above motivation scenario, such as age, diabetes type, BMI, cardiovascular risk factors, symptoms, dose type, and diagnosed criteria. The identified key concepts are considered as the main concepts of the ontology. Then, we identified key concepts to identify the sub-concepts and properties which describe the relationship between concepts. For this process, we used designed conceptual models (Figure 1, Figure 2 and Figure 3) to verify the identified concepts, sub-concepts and their relationships that are correctly identified according to the clinical guidelines. Furthermore, we used these conceptual models to validate the content of the proposed ontology. Table 1 demonstrates the generalization of identified key concepts with respect to the above motivation scenario.

We identified age as a key concept with respect to this scenario. When we consider the relationship between age and diabetes, according to the diabetes clinical guidelines, four age groups are related to the diagnosis and treatment of diabetes. Therefore, these age groups are identified as the sub-concepts of the age concept. The same procedure was used to identify the sub-concepts of the diabetes types concept. The instances related to the identified concepts are obtained using the generalization of the motivating scenario. For example, we studied clinical guidelines to identify the available screenings for diabetes. These screenings were identified as instances of the screening concept. We applied this procedure for all motivating scenarios to identify the core design elements of the ontology structure.

After identifying concepts, sub-concepts and instances, we have used designed conceptual models to identify the relationship between the concepts. For that, we analyzed how identified concepts are related to each other in the healthcare domain concerning diabetes. For example, when we consider the relationship between diabetes types and diagnosis criteria, each diabetes type has the specific diagnosed criteria and screening. In addition, there is a relationship between the diagnosis criteria and screening. We can formulate those relationships between concepts as follows (see Table 2).

Next, we formulated the identified concepts and properties according to the first-order logic (FOL). The concepts are represented using unary predicate and the properties are represented using binary predicates. Table 3 shows the representation of concepts and properties in FOL.

Then, formal competency questions (CQs) are formulated by using FOL terminologies. The following example shows the formulation of formal CQ with respect to informal CQ 1.


$x $c $d (OralAntibioticAgent($x)



˄ hasAdvantage($x,weight_loss)



˄ hasAdvantage($x, effective_glycemic_control)



˄ hasAdvantage($x, cardiovascular_protection)



˄ GlycemicControl($d)



˄ hasGlycemicControlStatus($d,poor)



˄ hasOralAntibioticAgent($d,biguanides)



˄ hasDrugCompound(biguanides,metformin)



˄ hasOralAntibioticAgent($d, sulfonylureas)



˄ DietPlan(dietplan1)



˄ hasStatus(dietplan1,good)



˄ AgeRange(46) ˄ BMIRange(Obesity)



˄ DiseaseType(type2)



˄ CVDRiskFactor(hypertension))


After identifying all the concepts, sub-concepts, properties, and criteria, the structure of the ontology was designed for all the motivation scenarios separately. We selected the Web Ontology Language (OWL) to implement the ontology. The actual representation of the domain knowledge sometimes contains the relationships between more than two individuals (concepts). Thus, the design of the ontology structure should be designed to make it suitable to be represented using OWL. For that, we used ontology design patterns to design the ontology. The ontology design patterns are the modeling solutions to the problems in ontology designing.

As an example, consider the relationship between the concepts *GlycemicTarget*, *Screening* and *Value*. Each glycemic target has a value range with respect to the particular screening. There are several screenings related to one glycemic target. Figure 4 shows the relationship among these three concepts.

This shows the n-ary relationship. Since the OWL cannot represent the n-ary relationship, we cannot interconnect the above three concepts as mentioned in the diagram. There is an ontology design pattern called Defining N-ary Relations on the Semantic Web which can be used to model the n-ary relations [27]. There are two types of designing strategy included in this design pattern. According to pattern 1, if one of the concepts in the relation is distinguished from other concepts, it is called the originator of the relationship. For example, the GlycemicTarget concept is the originator of the above relationship, because this relationship describes two properties of GlycemicTarget, which are related to screening and the corresponding value range. Therefore, the most important concept of this relation is the GlycemicTarget. It is the owner of this relationship. The new concept should be introduced to represent this kind of relationship according to pattern 1 [27]. Therefore, we introduced a new concept, namely *GlycemicCriteria*, to represent the relationship among those concepts, as shown in Figure 5.

Design pattern 2 related to n-ary relations is used when all concepts in a relationship have equal importance. Each concept plays a different role in the relationship and has a different context. For example, a treatment plan of diabetes management depends on several criteria, such as age, BMI, risk factors, and the results of the screening. Since all the factors have the same importance, we cannot identify the originator of the relationship. Design pattern 2 is suitable for this kind of situation, which introduces a new concept to represent the relationship with links to all participants. Therefore, we introduced a new concept, namely *ManagementPlan*, to link all concepts related to the treatment as shown in Figure 6.

Figure 7 shows the ontology structure to represent knowledge for motivation scenario 1. The management plan of diabetes management can be divided into three subgroups, such as glycemic control, diet plan, and exercise schedule. All three subgroups have different properties and common properties. Therefore, the relationship between the management plan and subgroups forms the hierarchical structure which is the relationship between the main concept and sub-concepts. The sub concepts acquire all the data and object properties of the main concept (super-concept). The main concept includes the common data and object properties, while the sub-concept includes specific data and object properties. If a database is used to represent the hierarchical structure, first we need to design an extended entity-relationship diagram that is very complex to design. Ontology provides a more accurate method to represent hierarchical structures. The designers can simply define the “IS-A” relationship between main concepts and sub-concepts (see Figure 7). Furthermore, in OWL, it is easy to implement a hierarchical structure. Therefore, ontology provides more expressive power than databases. Similarly, other concept models were designed for all the motivation scenarios by using ontology design patterns.

### 2.2. Ontological Clinical Decision Support System

#### 2.2.1. Ontology Implementation

Before implementing ontology, the selection of an ontology representation language and an implementation tool is very important. OWL-2 Description Logics (DL) was selected to implement the designed ontology as a representation language. It provides well-defined syntaxes and semantics, sufficient expressive power, and efficient reasoning support to represent ontology [28]. Protégé was selected as a representation tool because it is an open source, standalone application with an extensible architecture. It provides a user-friendly environment with an ontology editor, built-in reasoner, and a plugin library that adds more functionality to the environment compared to the other tools available for ontology implementation. Finally, the ontology was implemented using the concepts, relationships, and criteria identified in Section 2.1.3. Furthermore, the ontological knowledge base was obtained by populating the ontology with instances.

#### 2.2.2. Proposed Clinical Decision Support System

The main objective of this research is to introduce a more suitable approach to implementing a clinical decision support system based on the clinical guidelines that is suitable for use in a local context. So, we introduced a novel approach to develop an ontology based on the clinical guidelines. The next step was to implement a CDSS. For that, we used the developed ontological knowledge base described in Section 2.2.1. Then, the Jena Framework was used to implement the knowledge access layer of the proposed system, as we decided to develop the proposed system as a web-based application. Jena is a Java framework for building semantic web applications. It provides extensive Java libraries that handle RDF (Resource Description Framework), RDFS (RDF Schema), RDFa (RDF in Attributes), OWL (Web Ontology Language) and SPARQL (SPARQL Protocol and RDF Query Language) in line with published W3C recommendations. Jena includes a rule-based inference engine to perform reasoning based on OWL and RDFS ontologies, and a variety of storage strategies to store RDF triples in memory or on disk [29]. We implemented SPARQL queries to retrieve information/knowledge from the developed ontological knowledge base by using Jena Framework and then converted RDF formatted data into the plain java objects which can be used in a Java application. Representational state transfer application program interfaces (REST APIs) that enable the access of a developed program through the World Wide Web were implemented using the Spring-Boot Framework. Further, we implemented a backend service of the proposed system by combining the above frameworks. The user interfaces (frontend application) were developed using the Angular Framework. Finally, the proposed system was implemented by connecting backend service with the frontend application.

Figure 8 shows the web-based form to get the user inputs related to the motivation scenario with respect to the management of type 2 diabetes and the relevant recommendations that are obtained from the ontological knowledge base by using SPARQL query.

## 3. Results

The main outcome of the DSR is designed IT artifacts. In this research, we introduced two novel design artifacts: (1) conceptual models by using the BPMN 2.0 to map the clinical guidelines and (2) an ontology structure for representing domain knowledge based on the clinical guidelines. These models are the major contributions for the healthcare domain as well as the computer field. In this section, we discuss in detail how we validated and evaluated these models.

### 3.1. Validating Conceptual Models

The validation process was carried out to validate the conceptual models (Figure 1, Figure 2 and Figure 3) developed to map the clinical guidelines. For that, we interviewed health professionals using pre-defined questionnaires to check the validity of the conceptual models and clarify the ambiguities in the clinical guidelines. In the clinical guidelines, there are some conditions of a patient that need to be considered when deciding treatment plans. However, there is no method or reasoning to determine those conditions. Therefore, we included these ambiguities in the questionnaires to identify the current method or reasoning used to determine these conditions. Furthermore, we presented the initial design of the conceptual models based on clinical guidelines to obtain the feedback from the experts. We asked them whether we had correctly mapped the concepts in the clinical guidelines, does the designed conceptual model completely represent the clinical guidelines, and does it provide a clear idea about the healthcare process included in clinical guidelines, etc. After getting their feedback, we refined the conceptual models. We conducted several interviews to obtain the final version of the conceptual models. For example, the process of distinguishing between type 1 and type 2 diabetes is not stated clearly in the clinical guidelines. It is very important to distinguish between type 1 and type 2 diabetes when deciding the treatment plan. Therefore, we worked with domain experts to design the actual diabetes classification process in the conceptual model. Finally, all the conceptual models were validated and then refined based on the domain expert suggestions/comments.

### 3.2. Ontology Structure Evaluation—Internal Evaluation

Three types of internal evaluation were carried out in this research to evaluate the developed ontology. The pellet reasoner was used to check the consistency of the developed ontology. If inconsistency occurred in the ontology, the reasoner provided reasons for inconsistency and allowed developers to resolve it. Therefore, we ensured the consistency of the ontology by using a reasoner.

Next, we checked whether the pitfalls exist in the implemented ontology. For this, we used a freely available online service called OOPS! (Ontology Pitfall Scanner). OOPS! provides (semi-)automatic diagnosis of OWL ontologies using an online and evolving catalogue of pitfalls. Figure 9 shows the evaluation results of the implemented ontology on OOPS! It shows that there are no critical and important level pitfalls in the ontology. Thus, we conclude that the quality of the implemented ontology is high.

Further to this, we created SPARQL queries for each CQ and obtained the answers. Then, we compared those answers with the expected answers for corresponding CQs. Figure 10 shows a SPARQL query related to the first CQ and the answers obtained from the ontology.


PREFIX rdf: <http://www.w3.org/1999/02/22-rdf-syntax-ns#>



PREFIX owl: <http://www.w3.org/2002/07/owl#>



PREFIX rdfs: <http://www.w3.org/2000/01/rdf-schema#>



PREFIX xsd: <http://www.w3.org/2001/XMLSchema#>



PREFIX dia: <http://www.semanticweb.org/sajith/ontologies/2020/3/diabetes-management#>



SELECT ?antibiotic WHERE{



{?antibiotic dia:hasAdvantage dia:WeightLoss.}



{?antibiotic dia:has Advantage dia:effective_glycemic_control.}



{?antibiotic dia:hasAdvantage dia:CardiovascularProtection.}}


#### Expected Answer: SGLT2 Inhibitor

The answer generated by the ontological knowledge base is correct for first CQ. In the same way, we obtained the answers for all the CQs using SPARQL queries. By comparing the expected answers and obtained answers, we can conclude that all the answers obtained by the knowledge base are accurate. By doing so, the accuracy of the ontology and thus the conceptual model (which is the basis of the ontology) are measured.

### 3.3. Ontology Structure Evaluation—External Evaluation

External evaluation was done by conducting interviews with health professionals using evaluation forms. In this phase, the suggested answers by the ontological knowledge base for CQs were evaluated. All the suggested answers were accurate and well-addressed in the real-world situations. In addition, some additional suggestions were added by them to enhance the ability for decision support of the model. They further mentioned that other clinical conditions should be considered when deciding the treatment plan for a particular scenario. For example, if a patient has type 1 diabetes at a young age, then the parents of the patient should be educated about diet plans, hypoglycemia, and giving insulin. Therefore, these kinds of additional information should be added to the treatment plans to increase the quality of the diabetes management. By considering all the facts, we can conclude that the developed model suggests the accurate answers.

## 4. Discussion, Conclusions and Future Directions

Most clinical guidelines are usually presented in non-computer-interpretable narrative text or non-executable flow chart formats. Furthermore, some guidelines published by international associations that are not suitable to use in some local contexts, because of the economic background, lack of resources, and that fact that some laboratory tests are not available in these countries. The existing approaches introduced ontology-based methods to implement clinical decision support systems for diabetes patients. Zhang and the team introduced a semantic-based approach to the unified representation of knowledge in a healthcare domain and patient data for practical clinical decision-making applications [21]. They implemented a semantic healthcare knowledge base based on the HL7 reference information model, including an ontology to model the domain knowledge and patient data and an expression repository to encode clinical decision-making rules and queries. Due to the inherent complexity of the reference information model, more work is needed to understand and model the related classes and attributes. A more simplified model is needed to improve design time and increase time efficiency. The research work presented in [30] used an ontology-based model for the diagnosis and treatment of diabetes patients in remote healthcare systems. The main limitation of this method is that it takes a lot of time to make a decision, because it follows a guessing engine with several rules for logic. Furthermore, this research only focuses on application-level performance evaluation. Since this research study introduces an ontology-based methodology, the ontology evaluation is essential to verify the validity and the completeness of the ontology. Encoding for Medical Information SNOMED CT converts case-based text features into SCT IDs and was proposed to generate a semantic reuse mechanism [31]. These authors used SNOMED CT coding to encode event-based knowledge and applied it to the diabetes diagnostic event-based dataset. Coded concepts were used to build an ontology to represent the domain knowledge that could be used for diabetes diagnosis. However, their method does not include the concept of developing a complete ontology based on diabetes knowledge, covering only 75% of the ontology because it does not include symptoms, drug information, and laboratory tests that affect the decision marking. So, this method does not provide a viable solution for the accurate and efficient diagnosis of diabetes.

By considering all the facts, as a solution, we have come up with a novel approach to design a conceptual model based on the clinical guidelines which is simple; this approach can be used to represent the domain knowledge in context as well as to verify the content of the model using different validation and evaluation criteria. In this research, the conceptual models were constructed based on the identified diabetes clinical guidelines. The conceptual models are one of the main contributions of this research that provides greater understandability of the diabetes clinical guidelines by converting guidelines from text format to well-structured models (Figure 1, Figure 2 and Figure 3). The understandability of the diabetes management plans can be increased using these models. Then, the designed conceptual models were validated by interviewing domain experts. For that, the questionnaires were used to obtain feedback from domain experts and modified conceptual models according to their feedback. Next, the ontology structure was designed to represent this knowledge with the guidelines in the Gruninger and Fox method. The designed ontology structure was implemented by using OWL-2 DL and the Protégé tool. Finally, the evaluation was carried out in two phases: internal and external. The ontology was internally evaluated using SPARQL queries; the OOPS! tool was used to find the pitfalls in the ontology. For the external evaluation, domain experts were interviewed to obtain feedback/comments to the answers of the CQs. Finally, a web-based application (i.e., clinical decision support system) was implemented to demonstrate that the developed ontology and this application can be used to access the knowledge base for quality decisions. In this research, although we discussed our contributions with respect to the diabetes and Sri Lankan context, the contributions are not limited to this context, and can be applied for any disease or any context.

In the future, we plan to further work with more domain experts to increase the validity of the model. This will increase the quality of the model because the common agreement about the domain knowledge can be made among the domain experts, and this will lead to increasing the correctness of the content of the model. In addition to that, we decided to improve the implemented clinical decision support system that can be used in diabetes clinics in Sri Lanka, which gets patient data/information as an input and outputs the appropriate management plan according to the circumstances of the patient.

## Figures and Tables

**Figure 1 healthcare-08-00573-f001:**
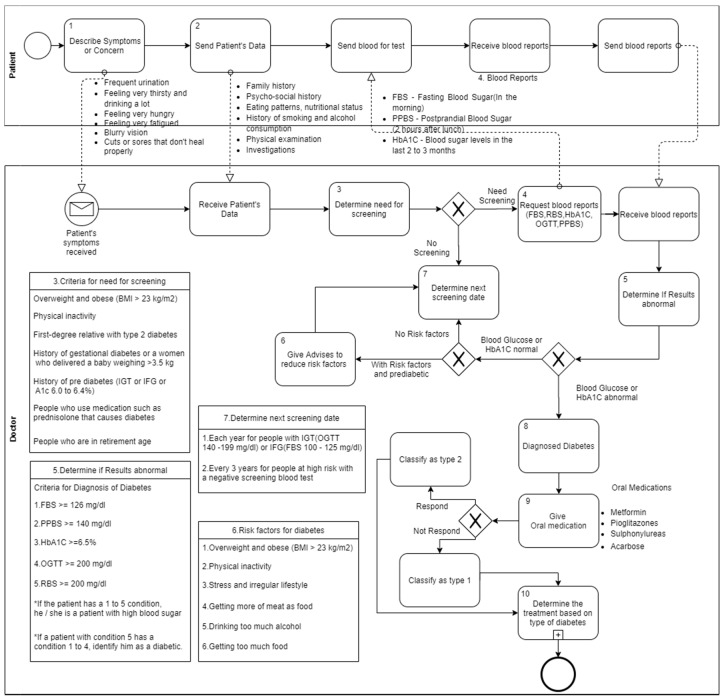
Conceptual model for diagnosing of type 1 and type 2 diabetes.

**Figure 2 healthcare-08-00573-f002:**
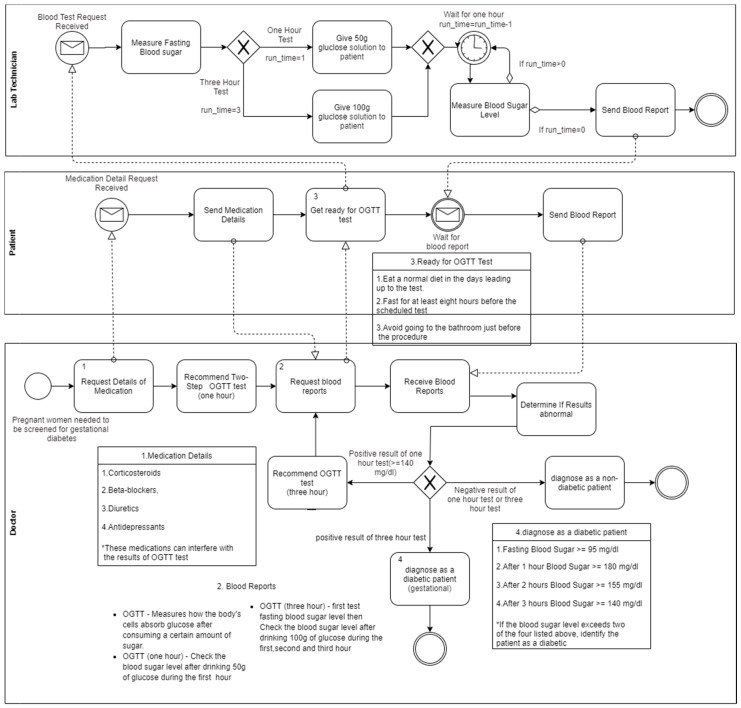
Conceptual model for diagnosing of gestational diabetes.

**Figure 3 healthcare-08-00573-f003:**
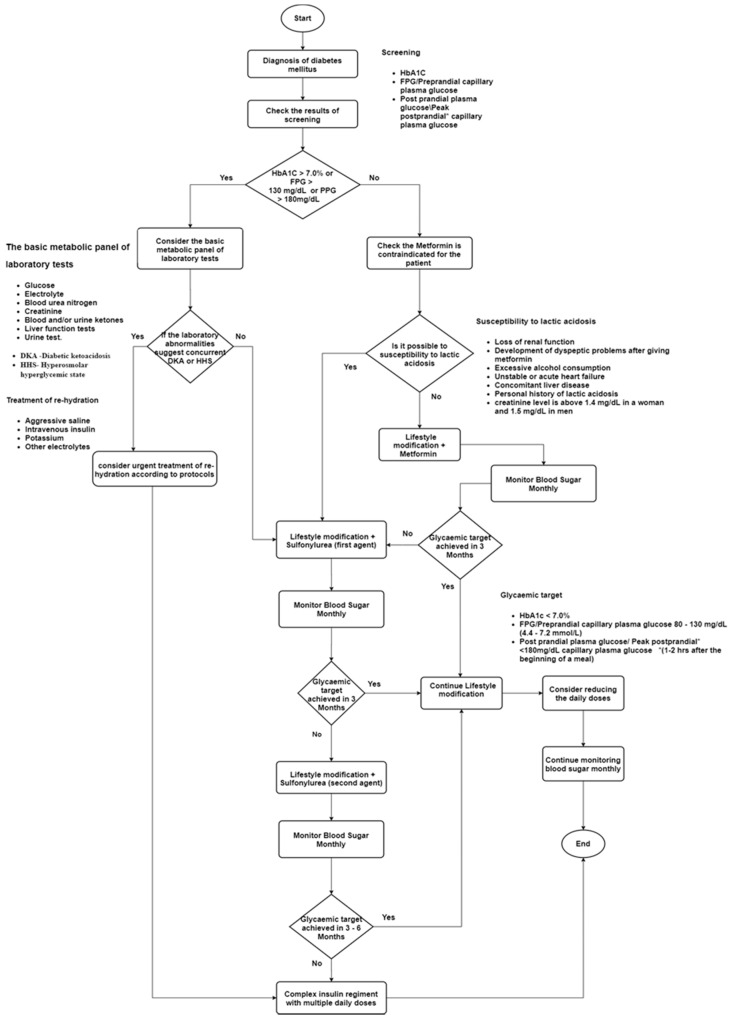
Conceptual model for non-insulin therapy.

**Figure 4 healthcare-08-00573-f004:**
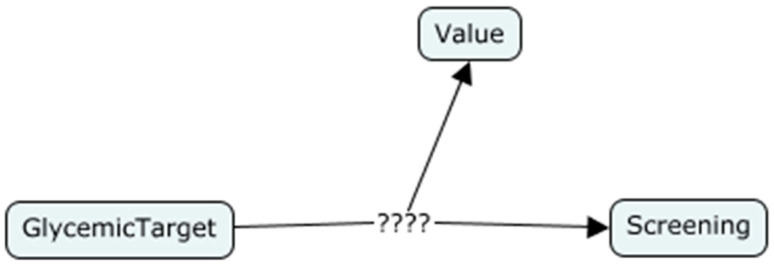
Before applying n-ary design pattern 1.

**Figure 5 healthcare-08-00573-f005:**
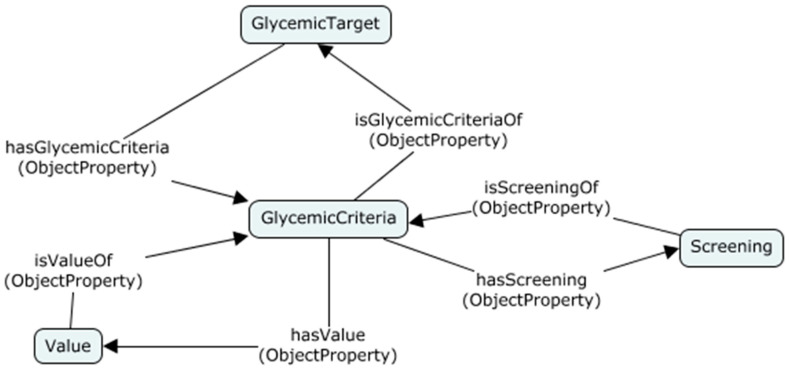
Applying n-ary design pattern 1.

**Figure 6 healthcare-08-00573-f006:**
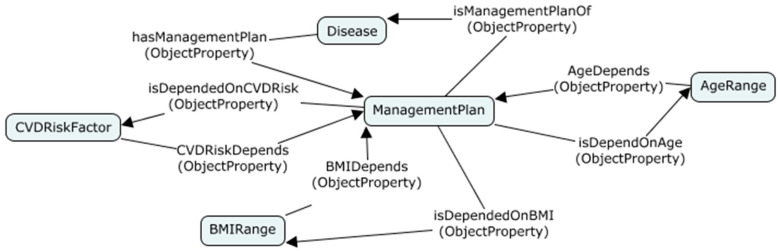
Applying n-ary design pattern 2.

**Figure 7 healthcare-08-00573-f007:**
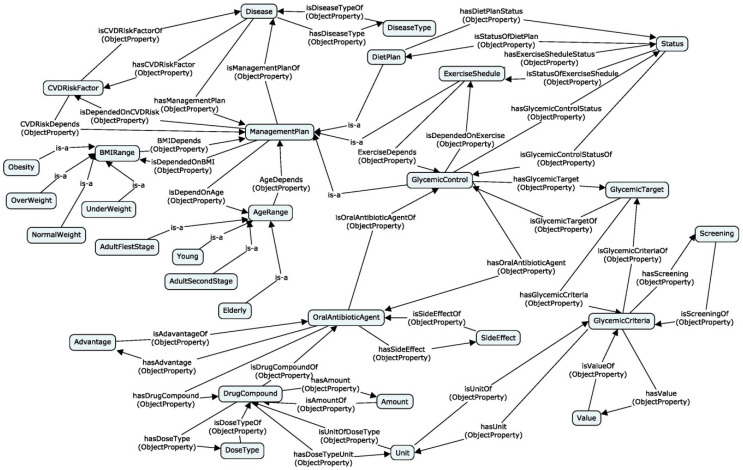
Ontology structure for motivation scenario 1.

**Figure 8 healthcare-08-00573-f008:**
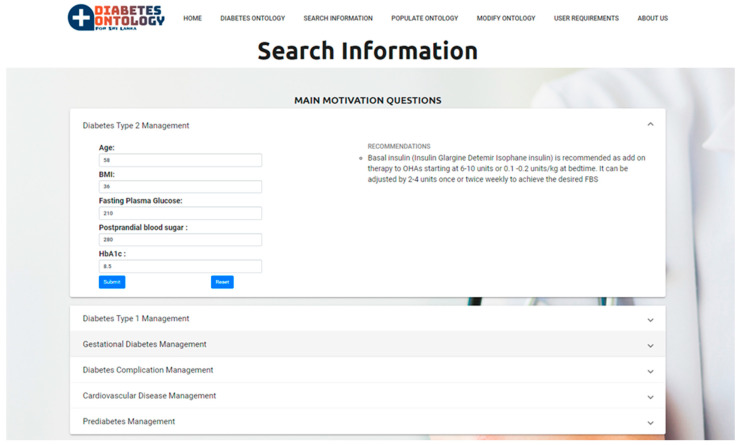
Web-based form related to type 2 diabetes management.

**Figure 9 healthcare-08-00573-f009:**
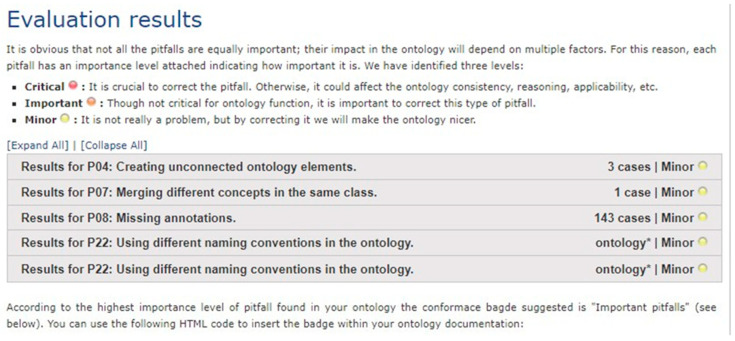
Evaluation results of implemented ontology on Ontology Pitfall Scanner (OOPS!).

**Figure 10 healthcare-08-00573-f010:**
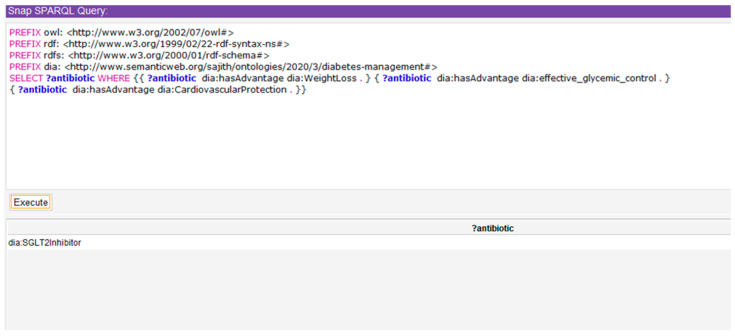
SPARQL query and Suggested answer for first competency question (CQ).

**Table 1 healthcare-08-00573-t001:** Generalization of key concepts.

Concept	Generalized Result
Age	Young <18 Adults first stage 18–40 Adults second stage 40–75 Elderly >75
Diabetes Types	Type 1 Type 2 Gestational
Cardiovascular risk factors	Hypertension Obesity Dyslipidemia Chronic Kidney disease Smoking Cerebrovascular disease
Screening	HbA1c FPG Pre-prandial capillary plasma glucose Postprandial plasma glucose

**Table 2 healthcare-08-00573-t002:** Identified relationships.

Concept (Source)	Relationship	Concept (Target)	Inverse Relationship
Disease Type	Has Diagnosis Criteria	Diagnosis Criteria	Is Diagnosis Criteria of
Diagnosis Criteria	Has Diagnosis Criteria Screening	Screening	Is Screening of Diagnosis Criteria

**Table 3 healthcare-08-00573-t003:** First order logic terminology.

Ontology Components	First Order Logic Terminology
Concept: Management Plan	ManagementPlan($y)
Property: has Oral Antibiotic Agent Concept (source): Management Plan Concept (Target): Oral Antibiotic Agent	ManagementPlan ($y) OralAntibioticAgent(biguanides) hasOralAntibioticAgent($y,biguanides)

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
