# Peer review of "An Ontological Clinical Decision Support System Based on Clinical Guidelines for Diabetes Patients in Sri Lanka"

_healthcare, 2020, doi:10.3390/healthcare8040573_

Round 1

Reviewer 1 Report

In this manuscript, the authors introduced an ontological model, which was designed for diagnosis and treatments of diabetes in developing countries, based on clinical guideline. Overall, this is interesting, and may be of clinical use.  

There are several minor concerns. 

-Please include the diagnosis and treatment plans for gestational diabetes in the design. In the Non-insulin therapy, is there any special attention that requires further attention for gestational diabetes? 

-What is the accuracy of the disease prognosis, diabetes detection, and medication prescriptions for the new model? 

-Compared with previous studies, what is the advantage of the current model? 

Reviewer 2 Report

The authors developed an ontological clinical decision support system based on clinical guidelines for diabetes patients in Sri Lanka. It is an interesting paper and provides a tool for medical informatics for applying in clinical medicine. From the points of view of physicians or patients, some comments below are for the authors to consider for revising the article. 

  1. There are many repeated or similar statements in the contents. It is a bit confusing about which part of the contents were for methods and which were for results. Suggesting that authors could write more precisely and reorganize the contents with following clear formats, including to reduce many contents of Introduction, even the authors tried to review the previous methods or technical issues, particularly for those that could be put into Methods or Discussion.
  2. For example, in Introduction, lines 52-54, 78-81, and 93-95, many statements about the necessity of ‘following clinical guidelines’ could be reduced. Also, the ‘main contribution’ of this study could be just kept at Discussion.
  3. Introduction, lines 69-71, the statement ‘We can easily overcome…’ may not be accepted by readers.
  4. It may be considered to modify the parts of ‘Materials and Methods’, ‘Designing and Implementation of The Proposed Models’, and ‘Results’ into two parts of ‘Materials and Method’ and ‘Results’, make the method and process more clearly and present results more logically.
  5. Section 2.3. Design Evaluation and 2.5. Research Rigor might be combined for those repeated statements.
  6. The paragraph, line 593-602, Results, the contents were more likely to be a summary to be put in Discussion. Also, the Discussion repeated much on the study process, lacking expanded views.
  7. In Figure 1, it does not give number 9 refer to line 359, page 9.
  8. For validating conceptual models, line 604-14, page 17, how many domain experts did the interview, what detailed questions in the pre-defined questionnaires were, and what exact feedbacks were from experts, these outcomes should be described. As there are many validation methods, not sure if the authors only conducted the content validation or facial validation here. It needs to clarify the internal/external evaluation or validation.
  9. The authors mentioned to ‘have used case sceneries’ from a reference article (25), line 398, page 11, and to ‘validate the content of the proposed ontology’, line 426-430, page 11-12. To follow the above comment, it would be good if the authors could examine the results from implementing the developed CDSS by using certain numbers of true diabetes cases from the experts’ institutes, to see how the consistency of outcomes between the clinicians and the CDSS, i.e., sensitivity, specificity, etc. This could be a validation process, if correct; and not clear if the authors mentioned this point at the end of the paper, line 683-89, page 20.
  10. Suggest that some results could be put in supplementary parts, or present precisely such as using tables for those on pages 12-13.
  11. For the references, please follow the journal requirement of the format, e.g., the internet assess date should be given.

Reviewer 3 Report

It is with great pleasure that I review this article.

In the abstract, the abbreviations should be avoided.

The introduction is too long. There are parts that can be cut.

Section 3 of the article is very well described and easy to understand. In the Figure 1 does not show the number 9, which is cited in the article on line 359.
Please confirm the citation [25] on line 398. They could explore the external evaluation a little better. How many clinicians? What parameters did the forms contain? Etc.

A conclusions section should be added to the article. This aspect is mentioned in line 182. But the last section presents the discussion.

Round 2

Reviewer 2 Report

    Thanks for the authors had taken time and effort to revise this manuscript. However, this paper was still written redundantly and not constructed well. Lots of statements might not be necessary, e.g., section 2.1. In addition, in line 358, page 11, the reference was cited differently from it in the last version.

Author Response

Response to Reviewer 2 Comments - 2nd round

Point 1: Thanks for the authors had taken time and effort to revise this manuscript. However, this paper was still written redundantly and not constructed well. Lots of statements might not be necessary, e.g., section 2.1.

Response 1: First of all, we would like to thank the reviewers for the very constructive reviews and helpful suggestions. We have resubmitted a new version of the article by considering all the suggestions and comments. We have responded each review comments.

For the 1st point, we have figured out that redundant statement in the section 2.1 as you mentioned. This explanation is aligned with the Design Science Research paradigm (section 2.1). We also feel that there are redundant statements. Therefore, we have removed this 2.1 Section (DSR Guideline) and reconstructed the paper. Further, we have removed lots of redundant statements according to your comment.

Point 2: In addition, in line 358, page 11, the reference was cited differently from it in the last version.

Response 2: It was a typing error in the initial version of this article which was corrected according to the reviewer’s comment in the previous round.

Further, we have restructured the article by focusing on the research design (clearly presented the conceptual model designing and implementation details in section 2); and evaluation methods and results were presented in section 3.1, 3.2 and 3.3 and it was improved appropriately.